# Spatial distribution and predictive factors of antenatal care in Burundi: A spatial and multilevel baseline analysis for the third burundian demographic and health survey

Emmanuel Barankanira[1], Arnaud Iradukunda [2,3,4,5]*, Nestor Ntakaburimvo[2,6], Willy Ahishakiye[7], Jean Claude Nsavyimana[7], Emmanuel Nene Odjidja[4,8,9]

1 Department of Natural Sciences, Burundi Higher Institute of Education, Bujumbura, Burundi, 2 Department of Statistics, Lake Tanganyika University, Bujumbura, Burundi, 3 Faculty of Medicine, University of Burundi, Bujumbura, Burundi, 4 Royal Society of Tropical Medicine and Hygiene, London, England, 5 Institute of Health Metrics and Evaluation, University of Washington, Washington, DC, United States of America, 6 Pathfinder International, Department of Monitoring, Evaluation & Learning, Bujumbura, Burundi, 7 Faculty of Fundamental Sciences, Armed Forces Avenue, Higher Institute of Military Cadres, Bujumbura, Burundi, 8 Department of Medicine, Monash University, Clayton, Australia, 9 Department of Monitoring and Evaluation, Global Community Engagement and Resilience Fund, Geneva, Suitzland

* arnaudiradukunda5@gmail.com

## Abstract

**Data Availability Statement:** The data underlying the results presented in the study are available

## Introduction

The use of antenatal care by pregnant women enables them to receive good pregnancy monitoring. This monitoring includes counseling, health instructions, examinations and tests to avoid pregnancy-related complications or death during childbirth. To avoid these complications, the World Health Organization (WHO) recommends at least four antenatal visits. Therefore, this study was conducted to identify predictive factors of antenatal care (ANC) among women aged 15 to 49 years and its spatial distribution in Burundi.

## Methods

We used data from the Second Burundi Demographic and Health Survey (DHS). A Spatial analysis of ANC prevalence and Mulitlevel logistic regressions of determinants factors of ANC with a medical doctor were done. The ANC prevalence was mapped by region and by province. In unsampled data points, a cluster based interpolation of ANC prevalence was done using the kernel method with an adaptive window. Predictive factors of ANC were assessed using Mulitlevel logistic regressions. The dependent variable was antenatal care with a medical doctor and the explanatory variables were place of residence, age, education level, religion, marital status of the woman, household wealth index and delivery place of the woman. Data processing and data analysis were done using using Quantum Geographic Information System (QGIS) and R software, version 3. 5. 0.

from (http://dhsprogram.com/data/availabledatasets.cfm).

**Funding:** The author(s) received no specific funding for this work.

**Competing interests:** The authors have declared that no competing interests exist.

**Abbreviations:** ANC, antenatal care; aOR, adjusted odds ratio; CI, confidence interval; DHS, demographic and health survey; EAs, Enumeration areas; HC, health center; OR, odds ratio; QGIS, quantum geographic information system; SDG, sustainable goals development.

## Results

The ANC prevalence varied from 0. 0 to 16. 2% with a median of 0. 5%. A highest predicted ANC prevalence was observed at Muyinga and Kirundo provinces' junction. Low prevalence was observed in several locations in all regions and provinces. The woman's education level and delivery place were significantly associated with antenatal care with a medical doctor.

## Conclusion

Globally, the ANC prevalence is low in Burundi. It varies across the country. There is an intra-regional or intra-provincial heterogeneity in term of ANC prevalence. Woman's education level and delivery place are significantly associated antenatal care. There is a need to consider these ANC disparities and factors in the design and strengthening of existing interventions aimed at increasing ANC visits.

## 1. Introduction

Worldwide, around 800 women die each day due to preventable causes related to childbirth and pregnancy [1] and 94% of them are in lower-middle income countries [2]. The use of antenatalcare (ANC) by pregnant women enables them to receive good pregnancy monitoring. Skilled care during pregnancy saves women and newborns lives [2, 3]. This care includes counseling, health instructions, examinations and tests to avoid pregnancy-related complications or death during childbirth [4]. To avoid these complications, the World Health Organization (WHO) recommends at least four ANC visits [5–7]. The most determinants include number of medical doctors at each community health center and distance to nearest hospital [8]. These visits are a good time to adopt safe behaviors and learn about parenting [9]. From conception to delivery, pregnancy monitoring promotes maternal and child health [5]. The third United Nations Sustainable Development Goal (SDG), in its targets 1 and 2, aims to reduce maternal mortality to less than 75 per 1000 live births and neonatal mortality to 12 per 1000 live births by 2030 worldwide [3, 10]. A systematic review of 23 countries in sub-Saharan Africa showed that the use of health care services by a pregnant woman has reduced neonatal mortality by 39% [5].

Evidence from the East African region, specifically from Kenya and Uganda, show that more educated women, wealthier and those living in cities were found to be more likely to use maternal health services,in cluding antenatal care [11, 12]. Then, even if there is a scarity of published evidence from East African countries (Burundi, South Soudan, etc. . .), maternal health service utilization remains persistently low [13]. Burundi experienced disparity in health care services access, including maternal services, due to the low number of health centers and health care delivers, particularly in rural areas [14]. Previous DHS reports showed that most of women attending antenatal clinics in Burundi, in many cases, during these clinics, essential tests are not performed and women do not receive important information about the risks of pregnancy.

A comprehensive understanding of ANC in term of spatial distribution and predictive factors is needed to inform policy and identify specifc programming interventions to achieve greater coverage of optimal ANC services and further improvements in maternal health care.

Therefore, this study aims to map the spatial distribution and to identify factors associated with ANC with a medical doctor among women aged 15–49 years in Burundi.

This study could provide, for public health decision-makers, information on where and why women consulted or not a medical doctor during pregnancy period. This information serves to raise the minds of health decision-makers by showing where (regions, provinces) there is a need for hospitals, health centers and qualified health care staff. It also serves to identify where awareness of antenatal and postnatal health care should be more raised.

## 2. Materials and methods

### 2.1. Data source, study design and setting

In this study, we used data from the Second Burundi Demographic and Health Survey (DHS) to serve as baseline for a similar study to be conducted on the third Burundi DHS in future. Moreover, no previous spatial analysis focused on prediction of ANC prevalence in unsampled areas have been conducted on DHS dataset in Burundi. Burundi shape files were downloaded from the website https://www.diva-gis.org.

Data were obtained from MEASURE DHS (Demographic and Health Surveys), a United Nations programme that assists developing countries in the collection and analysis of population, health and reproductive health data. The data relating to health centers came from the survey on the evaluation of the quality of services in health facilities in Burundi carried out in 2013 by the Ministry of Public Health and AIDS Control. Population data are demographic projections obtained from the Institute of Statistics and Economic Studies of Burundi in collaboration with the the National Institute of Public Health of Burundi under the ICF International technical assistance [15]. The survey was representative of the population.

### 2.2. Population and sampling procedure

The study population included all pregnant women from five years before the survey. The Second Burundi DHS uses a two-step stratified cluster sampling method in which sample households are selected in cluster or enumeration areas (EAs) [16]. At the first stage, 376 clusters (301 rural and 75 urban clusters) were randomly selected with a proportional probability to their size (number of households) from the 8,104 clusters of the 2008 General Census of Population and Housing. At the second stage, 24 households were allocated in each selected cluster, leading to a sample of 9024 households. The stratification was done according to province (there were 17 provinces in 2010) and place of residence. The clusters were distributed at the national level according to place of residence (Fig 1). Burundi has currently 18 provinces grouped into 5 health regions (Bujumbura-Mairie, West, South, North and Centre- East). The new province of Rumonge (which was a commune of the province of Bururi) is the result of the merger between three former communes of Bururi (Rumonge, Buyengero and Burambi) and two former communes of Bujumbura-Rural (Bugarama and Muhuta).

During the survey, 9389 women aged 15–49 years in all selected households and 4280 men aged 15–59 years in half of the selected households for women were successfully interviewed. For women with multiple births, we consider the data for the most recent pregnancy with the view of minimising recall bias [17]. Women who had not given birth in the last 5 years prior to the survey and women who had not consulted a health care provider were excluded from the study. Therefore, taking account a calculating and reporting effect sizes [18], the study sample size consists of 5063 Burundian women aged 15–49 years who reported at least one live birth in the five years preceding the survey. The **figure** below shows, at national level, the clusters distribution according to place of residence. The red dots are urban clusters and the green dots are rural clusters.

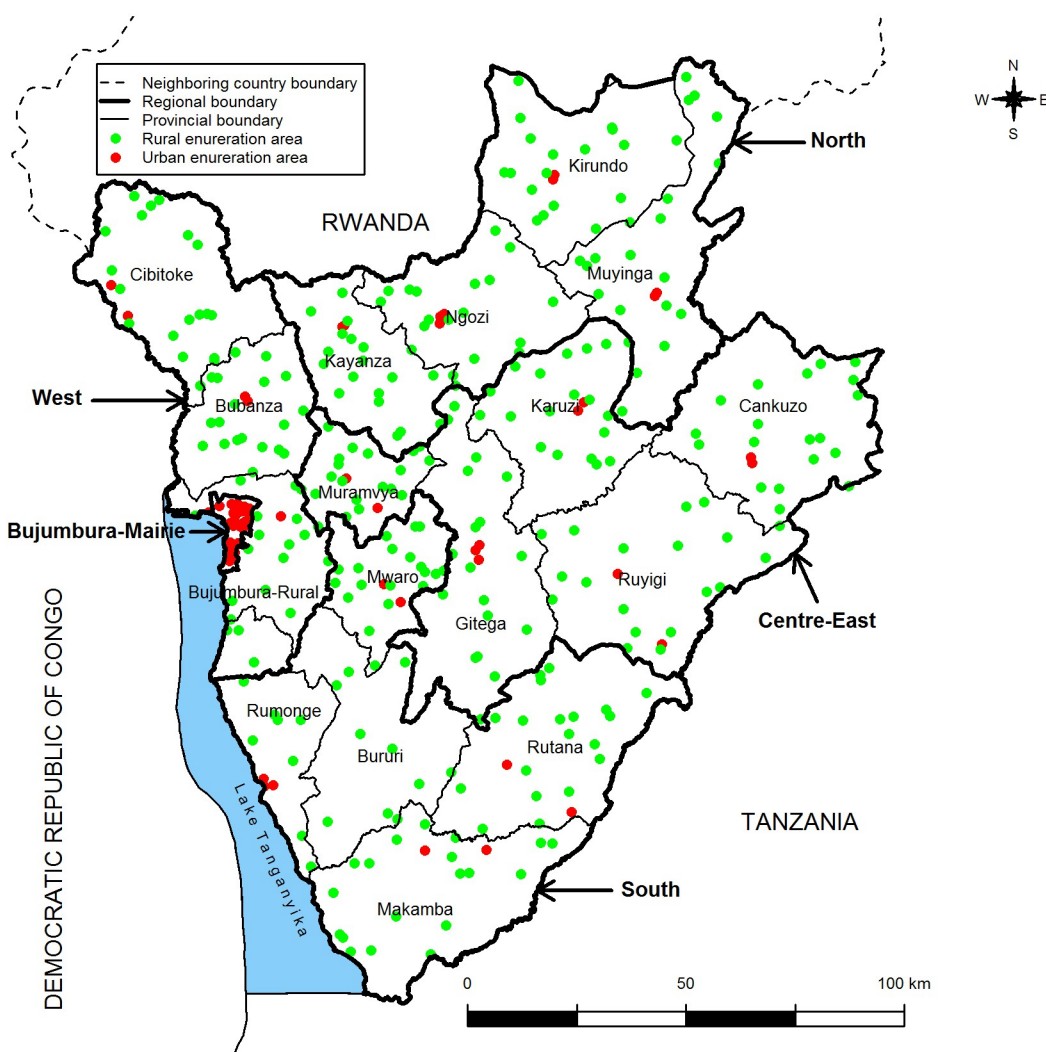

**Fig 1. Spatial distribution of clusters.**

### 2.3. Study variables

Based on theoretical and practical significance and the availability of the variables in the dataset, we considered both individual- and household-level factors in our study. According to the previous studies, women's education, education level of the woman, husband's education education level of her husband, place of residence, planned pregnancy, family income and household wealth index have been reported as determinants of antenatal care use [5, 19]. Several other studies have also shown that living in rural areas, low education level, low economic welfare quintile, low income, and being single predispose women to avoid using antenatal care services [20–22].

### 2.4. Individual-level factors

The explanatory variables of our study were individual level factors including woman's age (15–19 years, 20–24 years, 25–29, 30–34 years, 35–39 years, 45–49 years), woman's education

level (None, Primary, Secondary, Tertiary), marital status (single, married, living with a partner, widowed,divorced,separated), religion (catholic, protestant, muslim, other) [23–26].

## 2.5. Household-level factors

The household-level factors were place of residence (urban,rural), the household wealth index (Very poor, Poor, Medium, Rich and Very rich). In the context of Burundi, where socio anthropologic factors influence lifestyle, additional determinants such as delivery place (home, hospital, health center, other) have been included in independent variables [23–27].

## 2.6. Data management and analysis

Data was extracted from the Second Burundi DHS. Sample weighting was done to adjust for the disproportionate allocation of samples to strata and regions in the survey process and to restore representation. We created the province of Rumonge and health regions using Quantum Geographic Information System (QGIS), version 1. 8. 0 [28, 29]. Shapefiles were manipulated using QGIS. Figs 1 to 5 were created using maptools, prevR, lattice, KernSmooth,

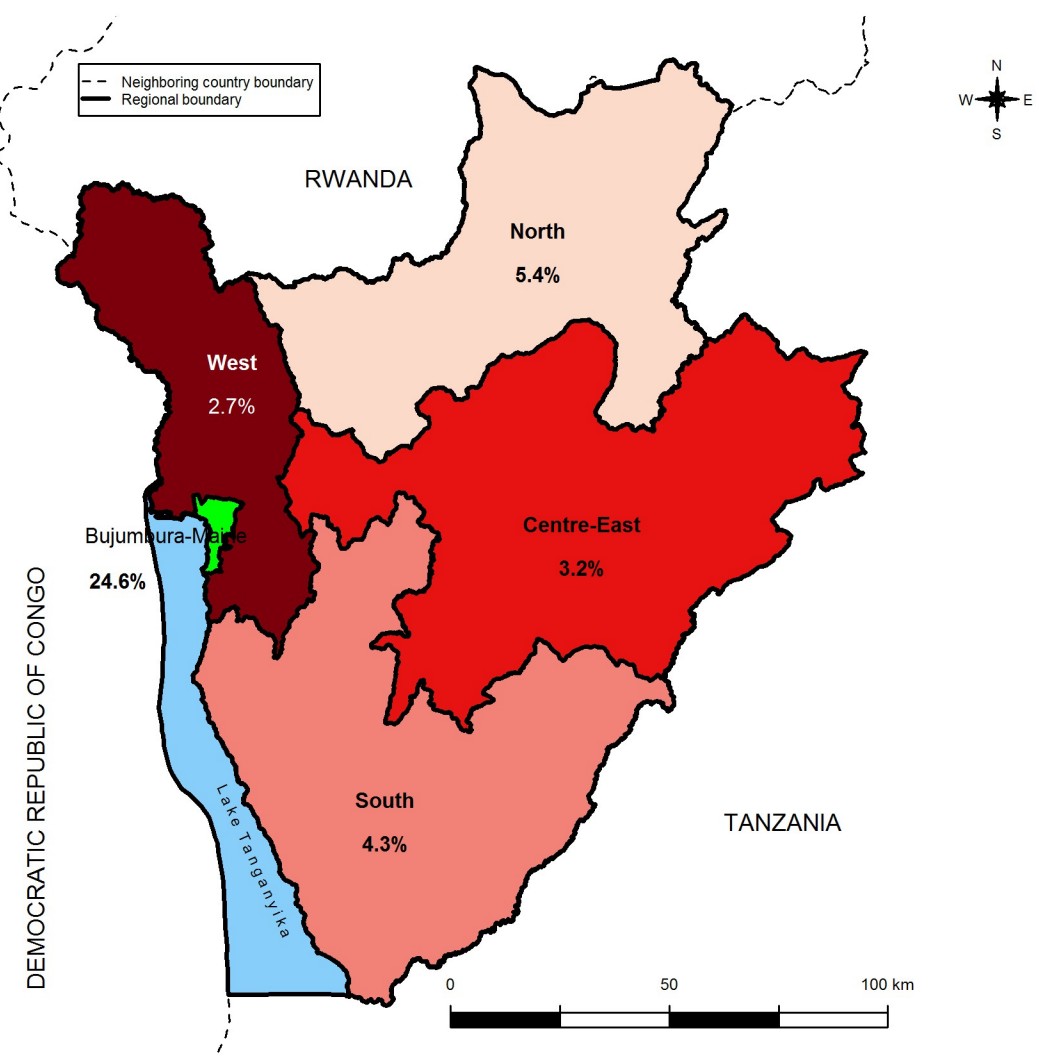

**Fig 2. Distribution of regional ANC prevalence for women aged 15–49 years.**

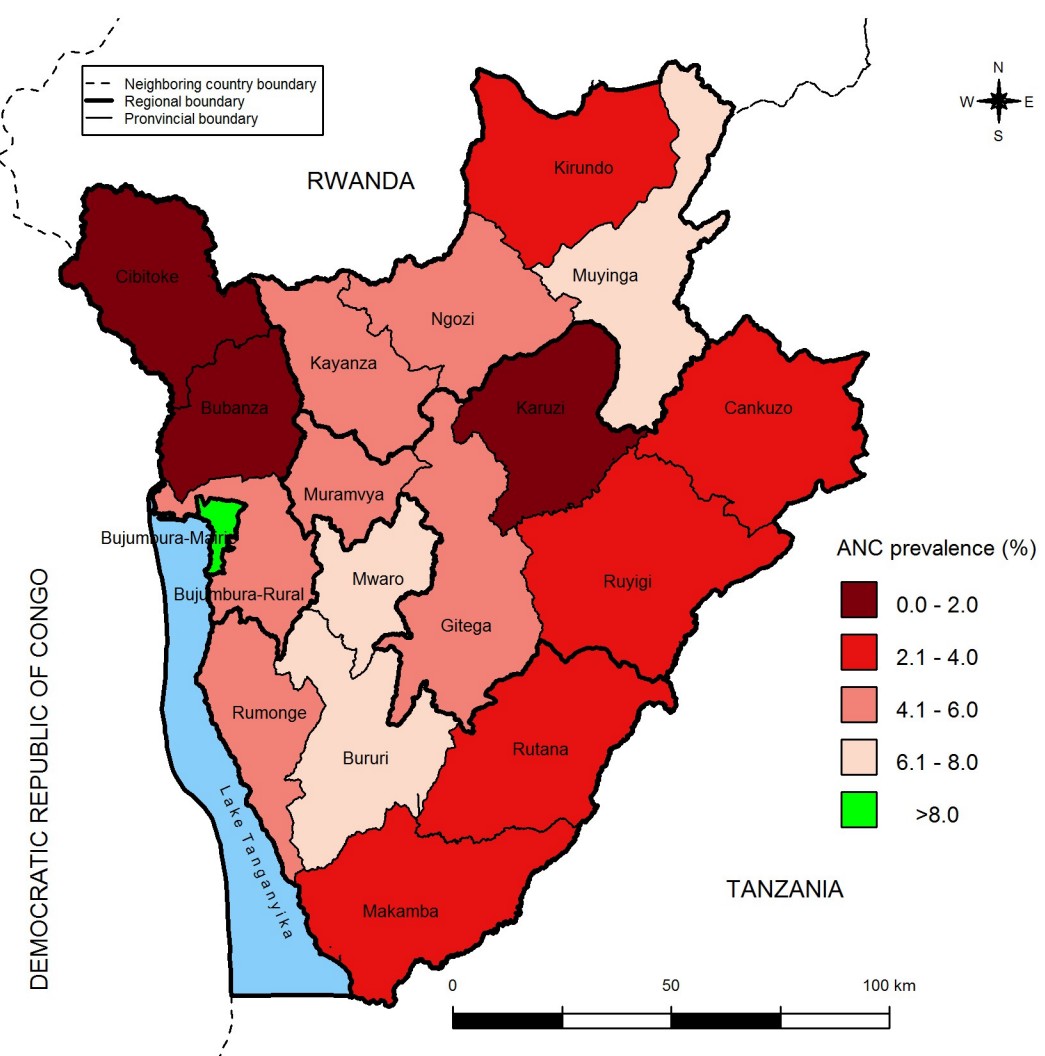

**Fig 3. Distribution of provincial ANC prevalence for women aged 15–49 years.**

graphics, sp and SDMTools R packages in R software version 3. 1. 3. We computed cluster-based (centroid of the cluster) ANC prevalence that we interpolated to unsampled data points.

For descriptive analysis,we firstly computed the total number of women, the number of women who had an ANC with a medical doctor, the number of women who did not have an ANC with a medical doctor. This variable allowed us to compute the ANC prevalence taking into account the sampling weights and using R software version 3. 5. 0 [30]. We secondly computed the ANC prevalence among women aged 15–49 years by province, region, place of residence, age groups, household wealth index, education level, religion, marital status and delivery place. Thirdly, we computed the ANC prevalence by cluster and socio-demographic characteristics. Lastly,we created a table of frequencies and percentages which showed the distribution of antenatal visits according to socio-economic characteristics.

For spatial analysis, we mapped ANC prevalence by region and province. In unsampled data points, we interpolated cluster-based ANC prevalence using the kernel method [31, 32]. For this, we created a regular interpolation grid (1000×1000) covering the whole country. We eliminated grid pixels that fall outside the country. This led to 85600 interpolation data points.

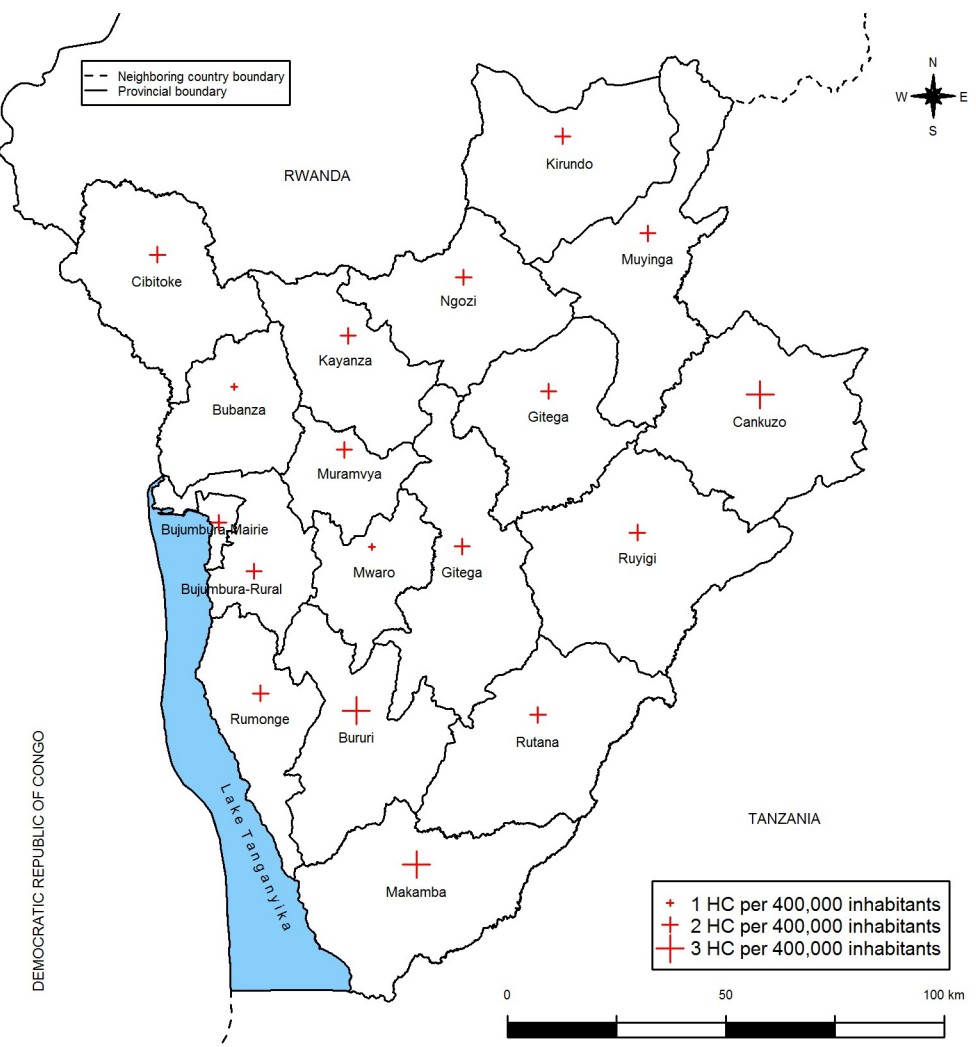

**Fig 4. Spatial distribution of health centers in Burundi.**

According to Larmarange J (2011), the modelling of the optimal value of $N$ (noted $N_0$) as a function of the observed national prevalence ($p$), the number of people tested ($n$) and the number of clusters surveyed ($g$) is given by the following equation [31, 33]:

$$N_0 = 2.688 \times n^{0.419} \times p^{-0.361} \times g^{0.087} - 91.011 \qquad (1)$$

We used $N_0 = 258$ (number of women in the search window) and a Gaussian kernel with an adaptive window.

## 2.7. Model specification and statistical modelling

We completed the descriptive and spatial analysis with a two-level multi-level binary logistic regression to assess the individual and household-level factors associated with ANC in Burundi. Model I contained individual-level variables alone and model II included household-level variables. The odds ratio and its corresponding 95% confidence intervals (CIs) were provided for models I & II. Using fixed effects binary logistic regression, we estimate the empirical

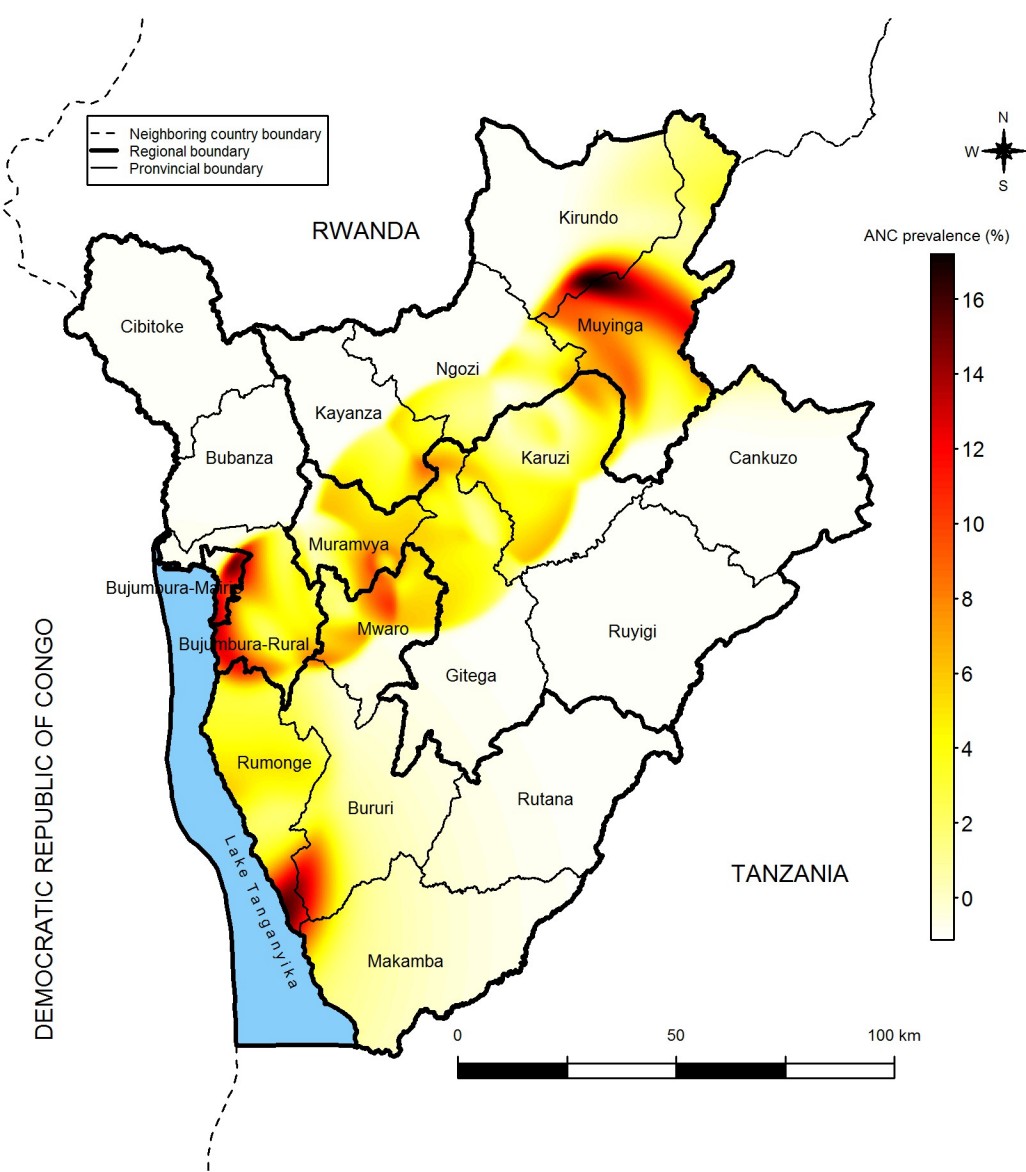

**Fig 5. Predicted ANC prevalence in unsampled data points.**

model to understand women's likelihood of utilising ANC provided by medical doctor, controlling for individual and household level characteristics:

$$M_i : \log\left[\frac{pi1}{pi0}\right] = \beta_0 + \beta_1 X_{ij} + \beta_2 Z_{ij} + \varepsilon_{ij}^1 \tag{2}$$

where p is the probability of consulting a medical doctor and ε a model error. Here, the dependent variable is the log odds that a woman i will choose alterative j relative to alternative 0, where 0 = non consultation with a medical doctor; and 1 = consultation with a medical doctor. Explanatory variabssles are grouped into two categories; namely individual-level factors represented by a standard vector of covariates X and household -level determinants corresponding to the standard vector of covariates Z. The dependent **variable is** a dummy variable

showing whether a woman consulted a medical doctor (yes,no). **An explanatory variable was considered** to be significantly associated with ANC when the overall p-value was less than 5%. Then, we created a table of results from univeriable analysis of associated factors with ANC. We introduced in the multivariable logistic model (full model) only the significant variables in univariable models. We also computed the adjusted odds ratios (aOR) obtained by exponentiating the parameters.

The 95% confidence intervals of aOR obtained by exponentiating the limits of confidence intervals of parameters. Thep-value which is the probability that the statistic of the test, under the null hypothesis, will be equal to or will exceed the estimated value was computed. We manually selected the final model using the backward step-by-step selection method with parsimony principle [34, 35]. The Bayesian Information Criterion (BIC) based on adjustment were used [36, 37]. The best model is one with low BIC value. Results of fitted model were presented in a table of aOR, CI and p value for each risk factor's category.

### 2.8. Patient and public involvement

This study used DHS datasets. No public health member or patient were involved in the design,analysis or reporting of this study.

## 3. Results

### 3.1. Spatial distribution of antenatal care

The proportion of women who consults a medical doctor varies across regions (Table 1). Infact, one woman over four women (25%) consults a medical doctor in Bujumbura-Mairie region. In others regions, less than 5% of women consults a Medical doctor. The table below shows the distribution of ANC prevalence by region.

The ANC prevalence is significantly higher in Bujumbura province than other provinces (Table 2). However, even if there is no significant difference among provincial ANC prevalences in same region, there is an intra-regional heterogeneity of ANC prevalence. The table below shows the pronvincial ANC prevalence distribution.

The ANC prevalence varies across regions (Fig 2). The highest ANC prevalence (24. 6%) is observed in Bujumbura-Mairie region where most of medical doctors and health centers are located. The lowest ANC prevalence (2. 7%) is observed in West region. Except Bujumbura mairie region, all other regions have less than 6% of ANC prevalence. The figure below shows the distribution of regional ANC prevalence for women aged 15–49 years.

The ANC prevalence decreases from Bujumbura-Mairie province to remote provinces to Bujumbura-Mairie (Fig 3). The ANC prevalence varies from 0. 4% in Karusi province to over 8. 0% in Bujumbura-Mairie. The Tanzania limitroph provinces (Makamba, Rutana, Ruyigi,

**Table 1. Distribution of the ANC prevalence by region.**

| Region | N$^*$ | N-$^*$ | N+$^*$ | % | 95% CI |
|---|---|---|---|---|---|
| **Bujumbura** | **268** | **202** | **66** | **24.6** | **[16.2; 33.1]** |
| North | 1606 | 1520 | 86 | 5.4 | [3.3; 7.5] |
| Centre-East | 1313 | 1271 | 42 | 3.2 | [1.7; 4.8] |
| West | 859 | 840 | 19 | 2.7 | [1.2; 4.3] |
| South | 1016 | 968 | 48 | 4.3 | [2.5; 6.1] |
| **Overall** | 5063 | 4801 | 262 | 5.2 | [4.1; 6.2] |

$^*$: weighted numbers are rounded; %: prevalence of ANC; CI: confidence interval

**Table 2. Distribution of antenatal care prevalence by province.**

| Province | N* | N-* | N+* | % | 95% CI |
|---|---|---|---|---|---|
| Bubanza | 246 | 243 | 3 | 1.2 | [0.0; 2.7] |
| **Bujumbura-Mairie** | **268** | **202** | **66** | **24.6** | **[16.2; 33.1]** |
| Bujumbura- Rural | 278 | 266 | 12 | 4.2 | [0.7; 7.7] |
| Bururi | 169 | 158 | 11 | 6.3 | [0.0; 13.1] |
| Cankuzo | 142 | 138 | 4 | 3.1 | [0.2; 5.9] |
| Cibitoke | 349 | 344 | 5 | 1.4 | [0.0; 3.0] |
| Gitega | 402 | 383 | 19 | 4.7 | [0.6; 8.9] |
| Karusi | 290 | 289 | 1 | 0.4 | [0.0; 1.1] |
| Kayanza | 345 | 327 | 18 | 5.1 | [1.2; 9.1] |
| Kirundo | 423 | 410 | 13 | 3.1 | [0.7; 5.5] |
| Makamba | 276 | 267 | 10 | 3.7 | [0.9; 6.0] |
| Muramvya | 200 | 191 | 9 | 4.4 | [1.4; 4.0] |
| Muyinga | 402 | 371 | 31 | 7.7 | [1.8; 13.5] |
| Mwaro | 157 | 146 | 11 | 6.9 | [5.3; 10.4] |
| Ngozi | 436 | 412 | 24 | 5.6 | [2.6; 8.6] |
| Rumonge | 218 | 207 | 11 | 5.1 | [1.6; 8.5] |
| Rutana | 196 | 191 | 5 | 2.8 | [0.0; 6.3] |
| Ruyigi | 265 | 256 | 9 | 3.2 | [0.4; 6.1] |
| **Overall** | 5063 | 4801 | 262 | 5.2 | [4.1; 6.2] |

*: weighted numbers are rounded; %: prevalence of antenatal care; CI: confidence interval

and Cankuzo) are with less than 4. 1% of ANC prevalence. The relatively high prevalence, between 6. 1% and 8%, is observed in Bururi and Mwaro provinces. The **figure** below shows the distribution of ANC prevalence for women aged 15–49 years by province.

The map bellow shows the spatial distribution of health centers in Burundi (Fig 4). The crosses are drawn according to the number of health centers (HC) per 400,000 inhabitants in each province in 2013.

Bubanza and Mwaro provinces have a low number of HC per 400,000 inhabitants whereas Cankuzo, Bururi and Makamba provinces have a high number of HC per 400,000 inhabitants. Except Bubanza provinces with both low ANC prevalence ($\leq$ 2. 0%) and low number of HC (1 HC for 400,000 inhabitant), provinces with low ANC prevalence are not necessarily those with a low number of health centers. In the similarly way, provinces with high ANC prevalence are not necessary those with a high number of HC. The figure below shows the predicted ANC prevalence at unsampled data points.

The predicted ANC prevalence in unsampled data points varies from 0 to 16. 2% with a median of 0. 5%. A high ANC prevalence is observed at Muyinga and Kirundo junction (Fig 5). Low predicted ANC prevalence is observed in several locations of the provinces, except in Rumonge province. The low predicted ANC prevalence is particularly observed in remote provinces of Bujumbura-Mairie (Ruyigi, Bubanza, Cankuzo, Bubanza and Cibitoke).

### 3.2. Socio-demographic characteristics

According to the household wealth index, the proportion of women who consult a medical doctor varies from 13% for women of wealthy families to less than 5% of women from other families (Table 3). The ANC prevalence increases with the household wealth index and the education level of the woman. Then, the ANC prevalence is very high among women with a

**Table 3. Distribution of antenatal care by socio-economic characteristics.**

| Characteristics | N* | N-* | N+* | % | 95% CI |
|---|---|---|---|---|---|
| **Place of residence** | | | | | |
| Urban | 440 | 358 | 82 | 18.7 | [12.8;24.6] |
| Rural | 4623 | 4443 | 180 | 3.9 | [2.9; 4.8] |
| **Age (years)** | | | | | |
| 15–19 | 158 | 150 | 8 | 5.3 | [1.4; 9.3] |
| 20–24 | 1093 | 1050 | 43 | 3.9 | [2.6; 5.2] |
| 25–29 | 1342 | 1260 | 82 | 6.1 | [4.4; 7.8] |
| 30–34 | 933 | 875 | 58 | 6.3 | [4.4; 8.1] |
| 35–39 | 892 | 849 | 43 | 4.8 | [3.2; 6.4] |
| 40–44 | 445 | 427 | 18 | 3.9 | [2.1; 5.8] |
| 45–49 | 200 | 190 | 10 | 4.7 | [1.6; 7.9] |
| **Wealth index quintile** | | | | | |
| Very poor | 1078 | 1047 | 31 | 2.9 | [1.5; 4.2] |
| Poor | 1093 | 1061 | 32 | 3.0 | [1.6; 4.3] |
| Medium | 1027 | 992 | 35 | 3.5 | [1.8; 5.1] |
| Rich | 972 | 929 | 43 | 4.4 | [2.7; 6.2] |
| Very rich | 893 | 772 | 121 | 13.4 | [10.1; 16.7] |
| **Educational level** | | | | | |
| None | 2666 | 2590 | 76 | 2.8 | [2.0; 3.7] |
| Primary | 2063 | 1962 | 101 | 4.9 | [3.5; 6.3] |
| Secondary | 301 | 237 | 64 | 21.2 | [15.2; 27.1] |
| Tertiary | 33 | 12 | 21 | 63.5 | [49.6; 77.1] |
| **Religion** | | | | | |
| Catholic | 3105 | 170 | 2935 | 5.5 | [4.2;6.7] |
| Protestant | 1570 | 67 | 1503 | 4.3 | [3.0; 5.5] |
| Muslim or other | 388 | 25 | 363 | 14.4 | [6.7; 22.0] |
| **Marital status** | | | | | |
| Single | 137 | 123 | 14 | 10.3 | [5.0; 15.5] |
| Married | 3120 | 2937 | 183 | 5.9 | [4.6; 7.1] |
| Living with a partner | 1385 | 1332 | 53 | 3.9 | [2.4; 5.3] |
| Widowed/ divorced/separated | 421 | 409 | 12 | 2.8 | [0.8; 4.7] |
| **Overall** | 5063 | 4801 | 262 | 5.2 | [4.1; 6.2] |

*: weighted numbers are rounded; %: prevalence of antenatal care; CI: confidence interval

high education level (64%) than women with no education level (3%). Comparatively to the woman's residence,women living in urban areas receive more ANC services (19%) from a medical doctor than those who live in rural areas(4%). The table below shows the antenatal care distributions according to the socio-economic characteristics.

### 3.3. Predictive factors of antenantal care

The Table 4 below shows factors associated with ANC which is a dependent variable for our study. It gives the odds ratio (OR), the 95% confidence interval (CI) and the p-value.

Multilevel logistic regresions models, for individual and household-level associated with ANC with a medical doctor in Burundi, show that the place of residence, wealth index quintile, education level, delivery place and marital status were significantly associated with antenatal care (Table 4).

**Table 4. Multi-level logistic regression models for individual and household-level factors associated with ANC with medical doctor in Burundi.**

| Characteristics | Model I | | | Model II | | |
|---|---|---|---|---|---|---|
| | OR | 95% CI | P-value | OR | 95% CI | P-value |
| *Age group (years)* | | | **0.126** | | | |
| 15–19 | 1.00 | | | | | |
| 20–24 | 0.73 | [0.31; 1.69] | 0.462 | | | |
| 25–29 | 1.16 | [0.50; 2.65] | 0.731 | | | |
| 30–34 | 1.19 | [0.54; 2.63] | 0.670 | | | |
| 35–39 | 0.90 | [0.39; 2.08] | 0.807 | | | |
| 40–44 | 0.73 | [0.28; 1.89] | 0.519 | | | |
| 45–49 | 0.88 | [0.30; 2.62] | 0.818 | | | |
| *Education level* | | | **<0.001** | | | |
| None | 1.00 | | | | | |
| Primary | 1.76 | [1.26; 2.46] | 0.001 | | | |
| Secondary | 9.15 | [5.74; 14.59] | <0.001 | | | |
| Tertiary | 59.38 | [30.68;114.90] | <0.001 | | | |
| *Religion* | | | **0.122** | | | |
| Catholic | 1.00 | | | | | |
| Protestant | 0.77 | [0.56; 1.07] | 0.121 | | | |
| Muslim or other | 1.22 | [0.71; 2.09] | 0.477 | | | |
| *Marital status* | | | **0.003** | | | |
| Single | 1.00 | | | | | |
| Married | 0.54 | [0.30; 0.99] | 0.046 | | | |
| Living with a partner | 0.35 | [0.18; 0.67] | 0.002 | | | |
| Widowed/divorced/separated | 0.25 | [0.09; 0.82] | 0.002 | | | |
| *Delivery place* | | | | | | **<0.001** |
| Home | | | | 1.00 | | |
| Hospital | | | | 4.13 | [0.18; 4.75] | <0.001 |
| Health care center | | | | 0.88 | [2.56; 6.97] | 0.656 |
| Other | | | | 1.51 | [0.48; 1.59] | 0.339 |
| *Place of residence* | | | | | | **<0.001** |
| Urban | | | | 1.00 | | |
| Rural | | | | 0.18 | [0.11; 0.28] | <0.001 |
| *Wealth index quintile* | | | | | | **<0.001** |
| Very poor | | | | 1.00 | | |
| Poor | | | | 1.03 | [0.53; 2.00] | 0.924 |
| Medium | | | | 1.21 | [0.63; 2.32] | 0.568 |
| Rich | | | | 1.57 | [0.85; 2.89] | 0.150 |
| Very rich | | | | 5.24 | [2.99; 9.20] | <0.001 |

OR: odds ratio; CI: confidence interval

The table below shows the fitted model.

Results shows finally that the factors associated with ANC with a medical doctor are the woman's education level and delivery place (Table 5).

The fitted model is written as:

$$M_f : \mathrm{logit}(p) = \beta_0 + \beta_1 \times \text{Education level} + \beta_2 \times \text{Delivery place} + \varepsilon \tag{3}$$

**Table 5. Fitted model.**

| Characteristics | aOR | 95% CI | P-value |
|---|---|---|---|
| **Education level** | | | **<0.001** |
| None | 1.00 | | |
| Primary | 1.54 | [1.10; 2.17] | 0.013 |
| Secondary | 6.44 | [4.01; 10.32] | <0.001 |
| Tertiary | 32.34 | [17.13; 61.05] | <0.001 |
| **Delivery place** | | | **<0.001** |
| Home | 1.00 | | |
| Hospital | 2.46 | [1.55; 3.93] | <0.001 |
| Health center | 0.73 | [0.40; 1.31] | 0.283 |
| Other | 1.45 | [0.60; 3.47] | 0.407 |

aOR: adjusted odds ratio; CI: confidence interval

where $\beta_0$, $\beta_1$ and $\beta_2$ are parameters that have been exponentiated in order to obtain adjusted Ors.

These results show that the ANC prevalence increases significantly with high education level, even after adjustment for the delivery place. Women who give birth in a hospital are 2. 46 times more likely to consult a medical doctor future pregnancy than women who give birth at home.

## 4. Discussion

This study aims to map the spatial distribution of ANC prevalence and examine factors associated with ANC with a medical doctor among women aged 15–49 years in Burundi. Findings show that ANC prevalence varies from one region (or province) to another and decreases from Bujumbura-Mairie(urban area) to the most remote regions and provinces(rural areas). These findings are consistent with findings of two recent studies, all conducted in Ethiopia, which showed a high ANC prevalence in urban areas (Addis Ababa) than rural areas [38, 39]. Regional antenatal care disparities in Burundi, especially when compared to capital of the country (Bujumbura-Mairie), are consistent with findings reported in another recent study [40]. These findings are also similar to those from two studies carried out in Indonesia and Nigeria which showed that ANC prevalence is higher in urban areas than rural areas [41, 42]. These findings could be due to the economic and urban profile of Bujumbura, a region with high-income comparatively to others regions. It could be due to the location of most health centers (hospitals, clinics, health center) and health care professionals in Bujumbura than remote regions. Moreover, those results are due to socio-economic differences and inequalities in access to health services between rural and urban areas in the country [43–45]. Therefore, geographical conditions show more variability among Burundian regions. In the line of findings of other studies, some areas are remote [46] and some others are quite difficult to reach due the limited available roads and public transportation [40]. There is no correlation between health centers and ANC prevalence, in terms of distribution, in most of provinces of Burundi. Then, most of regions are with low ANC prevalence. This could be explained by the shortage of medical doctors across the country in general and in the most remote areas of the capital city in particular.

The spatial analysis of predicted ANC prevalence in unsampled areas revealed variation across the country. High predicted prevalence were detected in Muyinga and Kirundo

junction, Bururi and Rumonge junction, Bujumbura-Mairie and Bujumbura Rural. These findings are in line with previous study conducted in Ethiopia and Indonesia [38, 47, 48]. In contrast, lower prevalence was observed in several locations of Burundi especially in remote provinces (Ruyigi, Bubanza, Cankuzo, Bubanza and Cibitoke). This disparity could be due the maternal health care inaccessibility and step-up in availability. These findings suggest to design effective intervention to improve ANC in low-prevalence areas of Burundi. Besides, a pocket of ANC prevalence was observed at the junction between Muyinga and Kirundo provinces. The predicted ANC prevalence is over eight percent in most of the parts of Muyinga province where observed ANC prevalence is high. The ANC prevalence increases with high education level. These findings are similar to those found in other studies carried out in Ethiopia and India which have reported disparities in antenatal care by place of residence (urban/rural), household wealth index and educational level of the woman [45, 47, 49]. Delivery place influences antenatal care use. In fact, as hospitals are located in urban areas where live women with a high education level, women wo give birth at these hospitals are more likely to consult a medical doctor during pregnancy period. Women who come for antenatal care are from not only the capital city of Bujumbura but also from the hills overlooking the capital in the province of Bujumbura-Rural and elsewhere. Moreover, women living in urban areas are more likely to consult a medical doctor than women living in rural areas [47]. Even though it is advisable to have at least 4 antenatal visits, some women do these consultations because they do not feel healthy and show a disengagement for postnatal care. Women often seek antenatal care based on advice from other women or health professionals who have already given birth. Failure to attend antenatal care increases the risk of maternal mortality, and attending antenatal care increases the likelihood of giving birth in health care facilities [50]. Women who give birth in a hospital setting are more likely to consult amedical doctor again than women who give birth at home, because women who visit a hospital usually have their own attending medical doctors. This is also justified by the fact thatmedical doctors are concentrated in hospitals in Burundi, and that these, especially private hospitals, are supposed to offer better quality care to pregnant women than other health centers.

Antenatal care in a health center can means that the woman is more likely to give birth in a health facility. Therefore, he have the chance to be assisted by a qualified health person and / or receive post-natal care.

Women's wealth index quintile, marital status, place of residence and educational level all favor the use of maternal health services [45, 51, 52]. Regarding the women's education level, the lack of knowledge influence young women in their maternal health services use: some of them consult lately and/or consult less than four times recommended.

In multi-level logistic regressions, factors associated with the antenatal with a medical doctor in our study are education level, marital status, delivery place, place of residence and wealth index quintile. Those factors are the keys to enhance the future success of ANC visits [53]. These finding are in line finding from a recent multilevel analysis of Ethiopian and Nigerian mini demographic health survey [54, 55]. Moreover, the fitted model shows that predicted factors of increased likelihood of making antenatal are the woman's education level and the delivery place. Considering the education level, women with primary level and women with secondary level are respectively 1. 54 and 6. 44 times more likely to consult a medical doctor than women with no education, while women with higher education level are 32. 34 times more likely to consult a medical doctor than women with no education level. Then, women who gives birth at hospital are 2. 46 times more likely to consult a medical doctor during pregnancy than women who gave birth at home.

These findings are consistent with previous studies conducted in Ethiopia and some East African countries which showed that women's education, health care accessibility and birth

order are major determinants of antenatal care [56, 57]. These findings are also in line with a recent study carried out in Ghana [58] which find that further education to reproductive-age women have significant role on ANC attendance.

The first strength of our study is that it combined a spatial analysis with an analysis of the factors associated with antenatal care among women aged 15–49 years in Burundi. The second strength is that this random sample is nationally representative because stratification was made by place of residence (urban/rural) and by province. Our results can be generalized to the entire Burundian population. The last strength is that we used dataset from the DHS which is based on a standardized questionnaire. Thecombination of the spatial analysis and the logistic regression model reinforces the relevance of our results. The interest of our study is that knowledge of the spatial distribution of ANC prevalence and factors associated with it will allow to know where awareness of maternal health needs to be more strengthened.

As a limitation, we did not use spatial logistic regression in modelling of antenatal care among women aged 15–49 years. We used a non-spatial logistic model as an additional analysis to the spatial analysis because the aim was not to take into account the spatial heterogeneity in the model. An additional study could detect spatial aggregates of high and low prevalence of antenatal care.

### 4.1. Policy implications

Findings from this study feed into the burundian context. Therefore, they could help public health policy decisions-makers to better identify where women are most affected by non-use of antenatal care among pregnant women and associated factors to help women adopt good behaviors related to pregnancy monitoring. These behaviors are influenced by sociocultural characteristics (level of education, place of residence, religion), sociodemographic characteristics (marital status, age and rank of pregnancy, surrounding), socioeconomic characteristics (standard of living, salary) and the geographical accessibility of health centers (distance to health care center, duration, transport means) [50, 59, 60]. These characteristics, in turn, influence pregnancy care, choice of delivery place and finally, antenatal and postnatal care visits [50]. To provide equitable access to antenatal care for all women, policy interventions could introduce free antenatal care into private health facilities to counteract access barriers to maternal health care.

### 5. Conclusion

Results of our study show that ANC prevalence is not the same across the country. It varies from region and province. However, there is intra-regional or intra- provincial heterogeneity in ANC prevalence. Predicting ANC prevalence at unsampled locations showed a pocket of ANC prevalence at the border of Kirundo and Muyinga provinces. Antenatal care was associated with woman's education level and delivery place. An analysis based on a Bayesian spatial approach could provide a good precision on the estimation of the parameters. A subsequent socio-anthropological study could identify women's perceptions of antenatal care, which would facilitate the early management of pregnancy.

### Acknowledgments

Authors are grateful to the Institute of Statistics and Economic Studies of Burundi, the Ministry of Public Health and AIDS Control (Burundi), and ICF International who conducted the Second Burundi DHS, and the DHS managers for access to datasets.

## Author Contributions

**Conceptualization:** Emmanuel Barankanira, Willy Ahishakiye.

**Data curation:** Emmanuel Barankanira, Arnaud Iradukunda, Nestor Ntakaburimvo.

**Formal analysis:** Emmanuel Barankanira, Arnaud Iradukunda, Willy Ahishakiye, Jean Claude Nsavyimana.

**Methodology:** Emmanuel Barankanira, Arnaud Iradukunda, Jean Claude Nsavyimana.

**Software:** Emmanuel Barankanira, Arnaud Iradukunda, Nestor Ntakaburimvo.

**Supervision:** Emmanuel Nene Odjidja.

**Validation:** Emmanuel Barankanira.

**Visualization:** Arnaud Iradukunda, Nestor Ntakaburimvo, Jean Claude Nsavyimana.

**Writing – original draft:** Emmanuel Barankanira, Arnaud Iradukunda.

**Writing – review & editing:** Emmanuel Barankanira, Arnaud Iradukunda, Nestor Ntakaburimvo, Emmanuel Nene Odjidja.

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
