## [Decision Letter · Decision Letter 0]

20 Sep 2022

PONE-D-22-21281Spatial distribution and predictive factors of antenatal care in Burundi: a multilevel analysis and spatial baseline for the third Burundian demographic health surveyPLOS ONE

Dear Dr. Iradukunda,

Thank you for submitting your manuscript to PLOS ONE. After careful consideration, we feel that it has merit but does not fully meet PLOS ONE’s publication criteria as it currently stands. Therefore, we invite you to submit a revised version of the manuscript that addresses the points raised during the review process.

Please respond to each comment made in detail. 

We look forward to receiving your revised manuscript.

Kind regards,

Mary Hamer Hodges, MBBS MRCP DSc

Academic Editor

PLOS ONE

Journal Requirements:

3. We note that Figures 1, 2, 3, 4 and 5 in your submission contain map images which may be copyrighted. All PLOS content is published under the Creative Commons Attribution License (CC BY 4.0), which means that the manuscript, images, and Supporting Information files will be freely available online, and any third party is permitted to access, download, copy, distribute, and use these materials in any way, even commercially, with proper attribution. For these reasons, we cannot publish previously copyrighted maps or satellite images created using proprietary data, such as Google software (Google Maps, Street View, and Earth). For more information, see our copyright guidelines: http://journals.plos.org/plosone/s/licenses-and-copyright.

a. You may seek permission from the original copyright holder of Figures 1, 2, 3, 4 and 5 to publish the content specifically under the CC BY 4.0 license.  

Reviewers' comments:

Reviewer's Responses to Questions

**Comments to the Author**

1. Is the manuscript technically sound, and do the data support the conclusions?

Reviewer #1: Yes

Reviewer #2: Yes

Reviewer #3: Yes

2. Has the statistical analysis been performed appropriately and rigorously? 

Reviewer #1: Yes

Reviewer #2: Yes

Reviewer #3: Yes

3. Have the authors made all data underlying the findings in their manuscript fully available?

Reviewer #1: Yes

Reviewer #2: Yes

Reviewer #3: No

4. Is the manuscript presented in an intelligible fashion and written in standard English?

Reviewer #1: Yes

Reviewer #2: No

Reviewer #3: Yes

5. Review Comments to the Author

Reviewer #1: Thank you for giving me this chance to review this article.

The manuscript is important and has merits. However, I have the following issues to be addressed by the authors;

1) Which BDHS did you exactly used? You said the second in the method part of the abstract section and the third in other sections. Please, be consistent on the phase of BDHS you used for this manuscript.

2) In the background of the abstract section you wrote this “Therefore, this study sought to understand the spatial distribution and predictive factors of antenatal care (ANC) among women aged 15 to 49 years with a medical doctor in Burundi.” The word understand, bolded one, is not measurable or it is not action verb. Hence, replace it by the other appropriate action verb.

3) To be in line with the PLOS ONE’s guideline, replace the word “background” by the word “Introduction”. Do this both in the abstract and in the Introduction section of the manuscript.

4) In the method section, you wrote it “Methods”. However, it is not in accordance with the PLOS ONE’s guideline. Replace it by “Materials and Methods”.

5) In the “study variables” part you didn’t write about the community level variables. It is necessary to include these variables, for you have done multilevel analysis.

6) Write about the methodology of multilevel analysis, the parameters and the number of models fitted in the data management and analysis part.

7) There are grammatical errors which need to be corrected. Example in the result section under the socio-demographic subsection you wrote this “The proportion of women who consults”….. make the verb consults plural i.e “The proportion of women who consult”….

8) Avoid italicizing words in some sections. Example in the Spatial distribution of ANC prevalence part of the result section you wrote the “figure below” in italics form.

9) In the predictive factors part of the result section you wrote this “Table4 gives the odds ratio (OR), the 95% confidence interval (CI) and the p-value associated with the explanatory variables when “antenatal care” with a medical doctor is the dependent variable .” The sentence doesn’t give any sense. Please, rewrite it!

10) You wrote unnecessary details about table 4. It seems discussion. Avoid explaining the result in the result section. You need simply present the table. Then, explain and discuss it in the discussion section.

Reviewer #2: The manuscript is technically sound and also the data supports the conclusion. Correct statistical analysis was used. The authors made all data underlying the findings in their manuscript fully available with out restriction.

Reviewer #3: I applaud the authors for adding significantly to the body of knowledge to improve maternal health in low-and-middle-income countries. Below I outline my comments

Introduction

The authors did not provide any background knowledge of maternal health in Burundi. For example,

What are the current ANC uptake rates in Burundi?

What are the underlying factors and determinants of ANC uptake in Burundi?

What is the health care environment in Burundi?

How does Burundi compare with other African countries?

Methods

What is “forimilar?”

I am unsure why the second Burundi DHS was used instead of the third. Please clarify and report the year the Burundi DHS was collected.

Please expand on the study variables. Report the variables alongside their multi-levels (individual, household, etc.). Without referencing the tables, readers should be able to tell what variables were measured.

Create subheadings, so the readers understand what and how it was done. See https://doi.org/10.3390/healthcare9101389, for example.

Results

Because the authors did not describe the variables in detail, it isn't easy to understand the result section. For example, the results discuss the prevalence in different regions. However, because this information was not introduced earlier, it becomes challenging to contextualize the results.

The authors report ORs of below 1.00 as an increase in likelihood. Check the paragraphs below in “Table 4. Univariate analysis of associated factors with ANC.”

Please rewrite the result section and the method section, so it flows. Describe the methodological sequence in the method section and report the same in the result section.

Discussion

The discussion should be contextualized to aid understanding. For example, it appears that the Bujumbura region is a high-income region, but without any context, this information can only be assumed. The authors should consider the international readership of this journal in their discussion.

When discussing the spatial results, I recommend the authors use generally acceptable languages. For example, “we found a high prevalence of ANC uptake in urban areas, such as Bujumbura.”

The paragraphs beginning with “The use of antenatal care….and antenatal care in a health center is a sign…” seems isolated and do not flow with the overall discussion and result. Which of the finding supports this paragraph?

Is this sentence reporting a finding from this current study “Due to lack of knowledge, young women make less use of maternal health services and consult late and then less than four times”?

Lastly, the authors should use an editing service to correct punctuation, spacing, and other minor grammatical errors.

6. PLOS authors have the option to publish the peer review history of their article (what does this mean?). If published, this will include your full peer review and any attached files.

Reviewer #1: **Yes: **Denekew Tenaw Anley

Reviewer #2: No

Reviewer #3: No

---

## [Author Response · Author response to Decision Letter 0]

15 Nov 2022

Dear Editor, 

Thank you for the further comments. Below is an overview of changes made to the manuscript and rebuttals (where applicable)

 Response to reviewer comments 

 Peer Reviewer 1 

Reviewer #1: Thank you for giving me this chance to review this article.

The manuscript is important and has merits. However, I have the following issues to be addressed by the authors; 

1) Which BDHS did you exactly used? You said the second in the method part of the abstract section and the third in other sections. Please, be consistent on the phase of BDHS you used for this manuscript.

Response 1: Thank you for your comments. We used the second demographic health demographic dataset. As we plan a similar study using the third DHS dataset, the most recent in Burundi, we used the second in order to have results of Burundi second DHS as baseline for the DHS the third data set. Moreover, no spatial analysis focused prediction of ANC prevalence in unsampled areas have been conducted on Burundi DHS dataset. 

2) In the background of the abstract section you wrote this “Therefore, this study sought to understand the spatial distribution and predictive factors of antenatal care (ANC) among women aged 15 to 49 years with a medical doctor in Burundi.” The word understand, bolded one, is not measurable or it is not action verb. Hence, replace it by the other appropriate action verb.

Response 2: Thank you for the comments. Changes have been made in the main manuscript. 

3) To be in line with the PLOS ONE’s guideline, replace the word “background” by the word “Introduction”. Do this both in the abstract and in the Introduction section of the manuscript.

Response 3: Thank you for the comments. Changes have been made in the main manuscript. 

4) In the method section, you wrote it “Methods”. However, it is not in accordance with the PLOS ONE’s guideline. Replace it by “Materials and Methods”.

Response 4: Thank you for the comments. Change have been made in the main manuscript. 

5) In the “study variables” part you didn’t write about the community level variables. It is necessary to include these variables, for you have done multilevel analysis. 

Response 5 : Thank you for the comment. Change have been made from title to the whole content of manuscript. We replaced “multilevel analysis” with “multivariate analysis.” 

6) Write about the methodology of multilevel analysis, the parameters and the number of models fitted in the data management and analysis part. 

Response 6: Thank you for the comment. Changes have been made.

7) There are grammatical errors which need to be corrected. Example in the result section under the socio-demographic subsection you wrote this “The proportion of women who consults”….. make the verb consults plural i.e “The proportion of women who consult”….

Response 7: Thank you for the comment. Change is made in the main manuscript.

8) Avoid italicizing words in some sections. Example in the spatial distribution of ANC prevalence part of the result section you wrote the “figure below” in italics form.

Response 8: Thank you for the comment. Change is made in the main manuscript.

9) In the predictive factors part of the result section you wrote this “Table4 gives the odds ratio (OR), the 95% confidence interval (CI) and the p-value associated with the explanatory variables when “antenatal care” with a medical doctor is the dependent variable.” The sentence doesn’t give any sense. Please, rewrite it!

Response 9: Thank you for the comment. Change is made in the main manuscript.

10) You wrote unnecessary details about table 4. It seems discussion. Avoid explaining the result in the result section. You need simply present the table. Then, explain and discuss it in the discussion section.

Response 10: Thank you for the comment. Change is made in the main manuscript.

Peer Reviewer 2

Reviewer #2: The manuscript is technically sound and also the data supports the conclusion. Correct statistical analysis was used. The authors made all data underlying the findings in their manuscript fully available without restriction 

Response: Thank you for the comments. We really appreciate it. 

Peer Reviewer 3

Reviewer #3: I applaud the authors for adding significantly to the body of knowledge to improve maternal health in low-and-middle-income countries. Below I outline my comments

Introduction

The authors did not provide any background knowledge of maternal health in Burundi. For example, What are the current ANC uptake rates in Burundi? What are the underlying factors and determinants of ANC uptake in Burundi? What is the health care environment in Burundi? How does Burundi compare with other African countries?

Response: Thank you for your comments. There is a scarity of published evidence from East African countries (especially from Burundi and South Soudan), maternal health service utilization remains persistently low. The health care environment in Burundi is that Burundi experienced disparity in health care services access, including maternal services, due to the low number of health centers and health care delivers, particularly in rural areas. More details have been made in the main manuscript: page 2: L19-25.

Methods

What is “forimilar?”

Response: Thank you for the comment. We correct this grammatical error by “for similar”

I am unsure why the second Burundi DHS was used instead of the third. Please clarify and report the year the Burundi DHS was collected.

Please expand on the study variables. Report the variables alongside their multi-levels (individual, household, etc.). Without referencing the tables, readers should be able to tell what variables were measured.

Create subheadings, so the readers understand what and how it was done. See https://doi.org/10.3390/healthcare9101389, for example.

Response: Thank you for your comments. We clearly mentioned all the reasons to use 2010 Burundi DHS instead the third Burundi DHS in the Materials and Methods section of this study (page 2:L40-45. This one was carried out in 2010. 

Results

Because the authors did not describe the variables in detail, it isn't easy to understand the result section. For example, the results discuss the prevalence in different regions. However, because this information was not introduced earlier, it becomes challenging to contextualize the results.

The authors report ORs of below 1.00 as an increase in likelihood. Check the paragraphs below in “Table 4. Univariate analysis of associated factors with ANC.”

Please rewrite the result section and the method section, so it flows. Describe the methodological sequence in the method section and report the same in the result section.

Response: Thank you for your comments. Changes have been made in the main manuscript: 

-We mentioned all variables categories. We also created two variables’ group of individual and community level.

-For the early introduction of the prevalence in different regions, a full description was already in the material and methods section 

-Results and methods sections are now rewritten: 

Discussion

The discussion should be contextualized to aid understanding. For example, it appears that the Bujumbura region is a high-income region, but without any context, this information can only be assumed. The authors should consider the international readership of this journal in their discussion.

When discussing the spatial results, I recommend the authors use generally acceptable languages. For example, “we found a high prevalence of ANC uptake in urban areas, such as Bujumbura.”

The paragraphs beginning with “The use of antenatal care….and antenatal care in a health center is a sign…” seems isolated and do not flow with the overall discussion and result. Which of the finding supports this paragraph?

Is this sentence reporting a finding from this current study “Due to lack of knowledge, young women make less use of maternal health services and consult late and then less than four times”?

Lastly, the authors should use an editing service to correct punctuation, spacing, and other minor grammatical errors.

Response: Thank you for your wonderful comments. We reviewed the whole discussion section. We rephrase all these sentences.________________________________________

Title:- Spatial distribution and predictive factors of antenatal care in Burundi: a multilevel analysis and spatial baseline for the third Burundian demographic health survey 

#Accept with Major revision General comments 

 The entire section of the manuscript contained missing/incomplete words. 

Response: Thank you for the comment. We revised all missing and incomplete words.

Abstract 

 The methods used to present findings (table, graphs, text, etc.) should be indicated in the abstract section.

Response: Thank you for the comment. Changes have been made in the main manuscript.

 The authors should describe the analysis techniques used to determine the determinant factors for ANC. The authors should describe the method used to report the effect size as well as the level of significance in the abstract section

Response: Thank you for the comment. Analysis techniques and methods used to determine the size effect have described. 

 Background:

 The authors should revise the background section Methods.

Response: Thank you for the comment. The background is revised. 

In the title you have stated the analysis as a multilevel analysis and spatial baseline for the third Burundian demographic health survey, but I have not seen any evidence in the method as well as in the result section that you have done a multilevel analysis, what was your intention to say a multilevel analysis?

Response: Thank you for the comment. The multilevel analysis have been stated in the methods section and done in the results section.

 In the method section, study variables part, the authors didn’t state about their dependent variable. Also the independent variables from previous studies should have to be merged and presented in a single paragraph.

Response: Thank you for the comment. The dependent variable is stated on Page 1.L34-35. Also, we considered both individual- and household-level factors in our study based on theoretical and practical significance and the availability of the variables in the dataset. 

 Results

 In the result section, on the sub section of Predictive factors of antenatal care, the authors have stated ANC with a medical doctor as a dependent variable. As per the DHS guide, women receiving antenatal care from a skilled provider for the most recent birth will be considered as having antenatal care. What is the reason behind choosing only ANC with a medical doctor? 

Response: Thank you for the comment. According for previous DHS reports (https://dhsprogram.com/pubs/pdf/FR253/FR253.pdf ), most of women attended antenatal clinics, but in many cases, during these clinics, essential tests are not performed and women do not receive important information about the risks of pregnancy. Those crucial information are supposed to be given by high qualified healthcare providers (a Medical doctors) prompt us to consider ANC with a Medical doctor 

In the result section, on the sub section of Predictive factors of antenatal care, replace the word univariate analysis by univariable analysis. Also replace the word multivariate analysis by multivariable analysis.

Response: Thank you for the comment. Changes have been made in the main manuscript. 

In the result section, education level and the delivery place were reported as a significant predictor of ANC. unless it is a previous delivery, how place of delivery affects ANC utilization? 

Response: Thank you for the comment. With delivery place variable, we mean here that the previous delivery place(s) play (s) a significant role in the ANC use and delivery place for the future pregnancy. 

Better to present the spatial distribution in a scientific way and report important results. For example, rather than doing the “Spatial distribution of health centers in Burundi” give emphasis for your outcome variable.

Response: Thank you for the comment. Spatial distribution of health centers in Burundi was presented jointly with the ration of population per health center. The disparities in term of HC should be one of the determinants of insufficient ANC use. Moreover, as we our outcome is about to have ANC with medical doctor. These medical doctors not found anywhere, but at their respective health Centers. Moreover we gave emphasis on our outcome variable in the results and discussion section.

---

## [Decision Letter · Decision Letter 1]

26 Dec 2022

Spatial Distribution and Predictive Factors of Antenatal Care in Burundi: A Spatial and Multilevel Baseline Analysis for the Third Burundian Demographic and  Health Survey

PONE-D-22-21281R1

Dear Dr. % arnaud Iradukunda%,

We’re pleased to inform you that your manuscript has been judged scientifically suitable for publication and will be formally accepted for publication once it meets all outstanding technical requirements.

Kind regards,

Mary Hamer Hodges, MBBS MRCP DSc

Academic Editor

PLOS ONE

Additional Editor Comments (optional):

Reviewers' comments:

Reviewer's Responses to Questions

**Comments to the Author**

1. If the authors have adequately addressed your comments raised in a previous round of review and you feel that this manuscript is now acceptable for publication, you may indicate that here to bypass the “Comments to the Author” section, enter your conflict of interest statement in the “Confidential to Editor” section, and submit your "Accept" recommendation.

Reviewer #1: All comments have been addressed

Reviewer #3: (No Response)

2. Is the manuscript technically sound, and do the data support the conclusions?

Reviewer #1: Yes

Reviewer #3: (No Response)

3. Has the statistical analysis been performed appropriately and rigorously? 

Reviewer #1: Yes

Reviewer #3: (No Response)

4. Have the authors made all data underlying the findings in their manuscript fully available?

Reviewer #1: Yes

Reviewer #3: (No Response)

5. Is the manuscript presented in an intelligible fashion and written in standard English?

Reviewer #1: Yes

Reviewer #3: (No Response)

6. Review Comments to the Author

Reviewer #1: (No Response)

Reviewer #3: (No Response)

7. PLOS authors have the option to publish the peer review history of their article (what does this mean?). If published, this will include your full peer review and any attached files.

Reviewer #1: **Yes: **Denekew Tenaw Anley

Reviewer #3: No

---

## [Editor Report · Acceptance letter]

3 Jan 2023

PONE-D-22-21281R1 

Spatial Distribution and Predictive Factors of Antenatal Care in Burundi: A Spatial and Multilevel Baseline Analysis for the Third Burundian Demographic and  Health Survey. 

Dear Dr. IRADUKUNDA:

I'm pleased to inform you that your manuscript has been deemed suitable for publication in PLOS ONE. Congratulations! Your manuscript is now with our production department. 

Kind regards, 

on behalf of

Prof. Mary Hamer Hodges 

Academic Editor

PLOS ONE